# Balancing Stability and Plasticity in Continual Learning:
# The readout-decomposition of activation change (RDAC) framework

## Abstract

Continual learning (CL) algorithms strive to acquire new knowledge while preserving prior information. However, this stability-plasticity trade-off remains a central challenge. This paper introduces a framework that dissects this trade-off, offering valuable insights into CL algorithms. The Readout-Decomposition of Activation Change (RDAC) framework first addresses the stability-plasticity dilemma and its relation to catastrophic forgetting. It relates learning-induced activation changes in the range of prior readouts to the degree of stability, and changes in the null space to the degree of plasticity. In deep non-linear networks tackling split-CIFAR-110 tasks, the framework clarifies the stability-plasticity trade-offs of the popular regularization algorithms Synaptic intelligence (SI), Elastic-weight consolidation (EWC), and learning without Forgetting (LwF), and replay based algorithms Gradient episodic memory (GEM), and data replay. GEM and data replay preserved both stability and plasticity, while SI, EWC, and LwF traded off plasticity for stability. The inability of the regularization algorithms to maintain plasticity was linked to them restricting the change of activations in the null space of the prior readout. Additionally, for one-hidden-layer linear neural networks, we derived a gradient decomposition algorithm to restrict activation change only in the range of the prior readouts, to maintain high stability while not further sacrificing plasticity. Results demonstrate that the algorithm maintained stability without significant plasticity loss. The RDAC framework not only informs the behavior of existing CL algorithms but also paves the way for novel CL approaches. Finally, it sheds light on the connection between learning-induced activation/representation changes and the stability-plasticity dilemma, also offering insights into representational drift in biological systems.

## 1 Introduction

Continual learning (CL) is a fundamental challenge in the field of neural networks, aiming to equip these systems with the ability to acquire new knowledge while not forgetting previously learned information, which is termed the stability-plasticity dilemma (Carpenter & Grossberg, 1987; Mermillod et al., 2013). In the common setting of gradient-based learning in neural networks, performance on prior tasks reduces as new tasks are learned i.e. stability is traded off for plasticity. This phenomenon is termed catastrophic forgetting (McCloskey & Cohen, 1989; French, 1999).

Much of such catastrophic forgetting has been linked to learning-induced changes that occur in the later layers of neural networks (Ramasesh et al., 2020; Kalb & Beyerer, 2022). Going a step further, recent work revealed that, during task-incremental learning (van de Ven et al., 2022), a large portion of the loss in task performance is not about the information relevant to the previous tasks being forgotten. Instead due to learning, the activations change, and the prior decision hyperplanes no longer align with the discriminative directions in those activations (Davari & Belilovsky, 2021; Anthes et al., 2023). Inspired by these results, we introduce a framework to study what aspects of such learning-induced activation changes are related to the performance on prior and future tasks i.e. the stability and plasticity during learning.

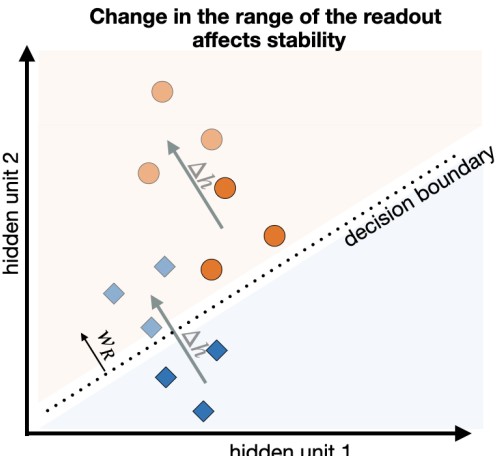 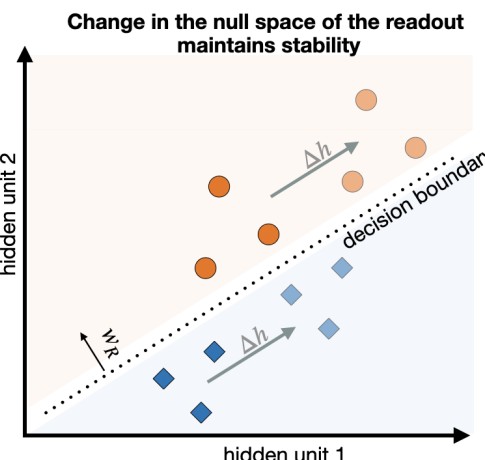

Figure 1: Conceptualization of the readout-decomposition of activity change. The readout for a task reads from a subspace of the pre-readout activation space. In the case of a two-dimensional activation space, and a binary readout, the hyperplane defines a line through the 2D space. (Left) Any gradient affecting this space in a way that changes activation patterns perpendicular to the hyperplane (the range of the readout) affects the mapping between hidden and readout activations, and is detrimental to stability. (Right) Any gradient affecting this space in a way that changes activation patterns parallel to the hyperplane (the null space of the readout) does not affect the mapping between hidden and readout activations. However, this change is useful for learning new tasks i.e. being plastic, while not disrupting stability.

We present the Readout-Decomposition of Activation Change (RDAC) framework. The central principle is as follows: Learning-induced activation changes within the range of a prior readout can lead to the task-relevant activations being misaligned with that readout, hampering stability. In contrast, changes in the null space do not affect stability, but they might reflect changes in the upstream weights due to the network learning new tasks. If so, restricting changes to the null space would make the network less capable of learning new tasks, hampering plasticity. This principle is exemplified in Figure 1.

The RDAC framework serves as a diagnostic tool to study the stability-plasticity trade-offs of CL algorithms. Moving beyond performance-based comparisons, we can probe the causes of those trade-offs by comparing them to the learning-induced activation changes in the range and null space of the prior readouts. In this work, we assessed the stability-plasticity trade-offs of popular CL algorithms - three regularization algorithms: Synaptic intelligence (SI; Zenke et al. (2017)), Elastic-weight consolidation (EWC; Kirkpatrick et al. (2017)), and Learning without forgetting (LwF; Li & Hoiem (2017)), and two rehearsal algorithms: Gradient episodic memory (GEM; Lopez-Paz & Ranzato (2017)), and data replay (Rebuffi et al., 2017).

Our results revealed that during learning the split-CIFAR-110 tasks, SI, EWC, and LwF traded off plasticity for stability, and restricted the activation changes in both the range and null space of prior readouts. Meanwhile, GEM and data replay maintained high stability and plasticity and surprisingly did not restrict activation changes in both the range and null space of prior readouts. We explain these observations in terms of the implementation details of those algorithms. Thus, we demonstrate the usefulness of RDAC in assessing the behavior of continual learning algorithms function at the level of the activations and the weights in the networks.

The RDAC framework can also guide the development of new continual learning algorithms, which shed further light on the relationship between the stability-plasticity trade-offs and the activation changes in the range and null space of the readouts. We demonstrate how we can translate the core principle of RDAC analytically into a gradient decomposition algorithm for simple one-hidden-layer linear networks (tested on split-MNIST tasks). We show that their stability-plasticity trade-offs are linked to the activation changes in the range and null space of the prior readout, and we can exert

full control over the trade-off with two parameters related to the range and null space of the readout. Thus, we validate the central principle of RDAC and show its value in building new CL algorithms.

In sum, the RDAC framework helps in assessing the stability-plasticity trade-offs in existing CL algorithms and paves the way for the development of new CL algorithms. By shedding light on these critical aspects of CL, this paper contributes to the ongoing efforts to characterize the complexities of continual learning in order to solve it.

## 2 RELATED WORK

In the landscape of research addressing the challenges of continual learning (CL), several noteworthy contributions have emerged that offer valuable insights into the stability-plasticity trade-off and related concepts (Parisi et al., 2019; De Lange et al., 2021; Mundt et al., 2023; Wang et al., 2023; Kim & Han, 2023). Most of these approaches compare the different CL algorithms in terms of their performance on previously-learned tasks as a function of learning new tasks and ablations on the various components of those algorithms. We seek to characterize the stability-plasticity trade-offs of the algorithms in terms of the changes in weights and prior tasks' activations. In addition to investigating how well algorithms solve the catastrophic forgetting problem, we place special emphasis on the algorithms' continued ability to learn new tasks i.e. their plasticity.

Recently, many groups have sought to understand the influence of learning-induced activation changes on stability and plasticity (Wang et al., 2021; Kong et al., 2022; Zhao et al., 2023; Yang et al., 2023). All these approaches analyze the gradient updates in the range of the space spanned by the activations for previous tasks, for each layer of the network. However, it is unclear whether the range of the readouts, which determines the outputs of the network, is equivalent to the range of the activations for the previous tasks. The range of the activations is most probably larger than the range of the readouts, and the pre-readout activation space is usually very high-dimensional compared to the readout. Also, prior work has shown that the activation space contains classification-orthogonal information (e.g. size and location of the object; Hong et al. (2016); Thorat et al. (2021)), which implies the activation space has more information than required by the readout. If the range of the activations is indeed larger, the space in which activations could change, while maintaining the previous tasks' performance, would seem smaller. This would suggest a reduced capacity for learning future tasks. To circumvent this potential issue, in the RDAC framework, we provide a different view on the link between the activation change and the stability-plasticity trade-off, by directly focusing on the range of the readouts.

## 3 THE READOUT-DECOMPOSITION OF ACTIVATION CHANGE FRAMEWORK

In the RDAC framework, we link the performance-based stability-plasticity trade-offs of continual learning algorithms to how they constrain learning. We study how learning-based changes in prior task activations, in the range and null space of the prior readouts, are linked to stability and plasticity.

In task-incremental continual learning, a network has to sequentially learn a set of $n$ task mappings $\{\mathbf{x}^{\mathbf{k}} \rightarrow \mathbf{o}^{\mathbf{k}}\}_{k \leq n}$. The network's weights are shared across tasks, except for the readouts, $\{\mathbf{W}_{\mathbf{R}^{\mathbf{k}}}\}_{k \leq n}$, which are task-specific. For a given task $k < m$, consider the pre-readout activations $\mathbf{h}^{\mathbf{k}}$ such that $\mathbf{o}^{\mathbf{k}} = \sigma(\mathbf{h}^{\mathbf{k}} \mathbf{W}_{\mathbf{R}^{\mathbf{k}}}^{\top} + \mathbf{b}_{\mathbf{R}^{\mathbf{k}}}^{\top})$, $\sigma$ is a non-linear activation function and $b$ is the readout bias. While learning on the new task $m$, we can decompose the change in activations, $\mathbf{\Delta h}^{\mathbf{k}}$ as $\mathbf{\Delta h}^{\mathbf{k}} \mathbf{C} \mathbf{C}^{\top} + \mathbf{\Delta h}^{\mathbf{k}} \mathbf{N} \mathbf{N}^{\top}$, where $\mathbf{C}$ and $\mathbf{N}$ are the range and null-space matrices of $\mathbf{W}_{\mathbf{R}^{\mathbf{k}}}$, and $\mathbf{C} \mathbf{C}^{\top}$ and $\mathbf{N} \mathbf{N}^{\top}$ are the corresponding projection matrices. These two components can provide us insights into the stability-plasticity trade-offs of a given learning algorithm.

First, changes in the activations in the range of the readout ($\mathbf{\Delta h}^{\mathbf{k}} \mathbf{C} \mathbf{C}^{\top}$) could disrupt stability as $\mathbf{\Delta h}^{\mathbf{k}} \mathbf{C} \mathbf{C}^{\mathbf{T}} \mathbf{W}_{\mathbf{R}^{\mathbf{k}}} = \mathbf{\Delta h}^{\mathbf{k}} \mathbf{W}_{\mathbf{R}^{\mathbf{k}}}$. A naive algorithm could preserve stability by ensuring that changes in activations do not happen in the range of the readouts. However, if the non-linearity $\sigma$ performs a many-to-one mapping, it can effectively negate the influence of those "degenerate" activation changes on the output. Smarter algorithms could change activations in the range of the readouts, even though their performance on previous tasks, i.e. stability, would be unchanged. In this case, the change of activations in the "effective range" (of the readout plus the further non-linearities),

would still be predicted to be zero. However, estimating effective range analytically is non-trivial, so this is harder to analyze.

Second, changes in the activations in the null space of the readout ($\mathbf{\Delta h^k N N^\top}$) have no effect on stability, as $\mathbf{\Delta h^k N N^T W_{R^k}^\top = 0}$. However, there could be a link between the changes in null space and learning on new tasks. As seen in previous work (Davari & Belilovsky, 2021; Anthes et al., 2023), without any learning constraints, changes in the prior tasks' activations are observed during learning. To maintain stability there is no incentive to reduce the activation change in the readout null space. A reduction in those changes in the null space could instead signal a reduced degree of learning on new tasks. In contrast, non-linearities upstream to $\mathbf{h^k}$ could perform many-to-one mapping in a way that weight changes do not get reflected in the changes in the activations in the null space, even if the changes in the range are clamped by a stability-preserving algorithm. Whether such many-to-one mappings occur in practice is an empirical question.

Given the above considerations, in RDAC we consider the following **Cases** to diagnose the stability-plasticity trade-off by considering the decomposed activation changes during task-incremental learning with a given continual learning algorithm:

1. **Stability** (preserved), **range clamping** (on): Aligns with the central principle of RDAC.
2. **Stability** (preserved), **range clamping** (off): The algorithm can access the effective range of the readout and the loss function combined, and/or the algorithm can access information related to previous data to encourage backward transfer.
3. **Stability** (hampered), **range clamping** (on): This is impossible.
4. **Stability** (hampered), **range clamping** (off): One way to gain stability is to modify the algorithm to lead to range clamping.
5. **Plasticity** (preserved), **null space clamping** (on): The upstream non-linearities create degenerate solutions for prior task mappings such that further learning can continue.
6. **Plasticity** (preserved), **null space clamping** (off): Aligns with central principle of RDAC.
7. **Plasticity** (hampered), **null space clamping** (on): The algorithm might be trading-off plasticity by clamping null space. We need to peer into the details of the algorithm to ascertain if the null space clamping could be removed.
8. **Plasticity** (hampered), **null space clamping** (off): If the null space is large enough, the algorithm has the capacity to learn but does not due to other causes, e.g. high learning rate.

Thus, monitoring these two components of the activation changes can provide us with valuable insights into the inner workings of continual learning algorithms.

## 4    ASSESSING EXISTING CONTINUAL LEARNING ALGORITHMS WITH RDAC

Using the RDAC framework, we present an assessment of how a variety of continual learning algorithms allow changes in the activations and how those changes relate to the observed performance-based stability-plasticity trade-offs. First, we outline the network architecture and task setting. Then, we present the results comparing the different learning algorithms: three regularization methods - Synaptic Intelligence (SI; Zenke et al. (2017)), EWC, and Learning without Forgetting (LwF; Li & Hoiem (2017)) - and two rehearsal methods - Gradient Episodic Memory (GEM; Lopez-Paz & Ranzato (2017)) and data replay[1](Rebuffi et al., 2017). In addition to them being popular, these algorithms were chosen due to the availability of their implementations through the Avalanche library (Carta et al., 2023).

### 4.1    NETWORK AND TASK

We considered a VGG-like neural network, similar to the one used in Zenke et al. (2017), which maps CIFAR images (Krizhevsky et al., 2009) to their classes (The exact architecture and optimization procedure is described in Appendix A.1.1). We considered the split-CIFAR-110 setting, where

---

[1]In data replay and LwF, contrary to the usual setting, we fixed readouts after training on a task and allowed the rest of the network to train further. This was done because in all the other methods, the prior readouts are frozen.

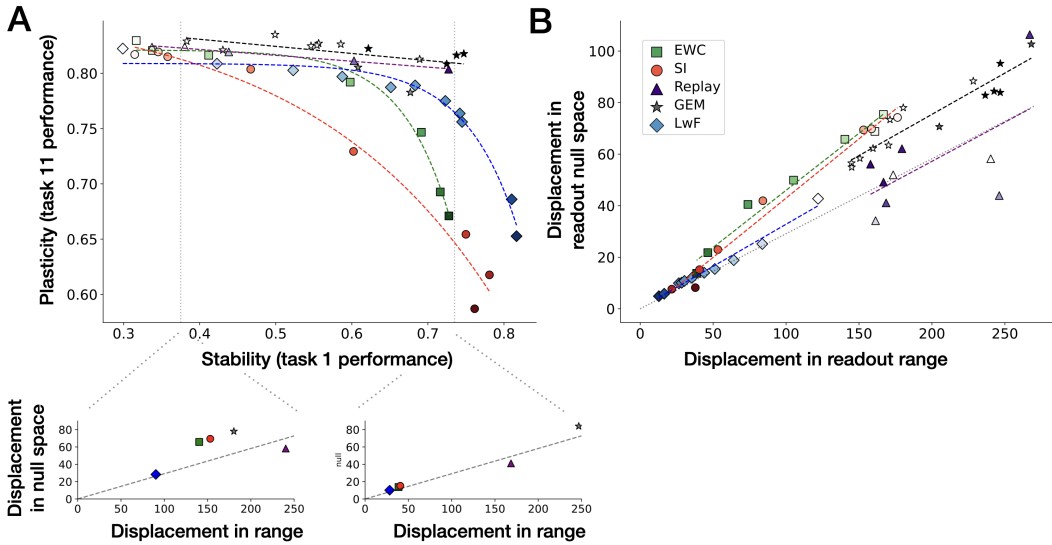

Figure 2: Continual learning in deep non-linear neural networks learning to classify CIFAR images. (A) With increasing regularization strength, replay buffer size, and memory strength (indicated with darkening hues), the stability of all the algorithms increases. However, the rehearsal algorithms, GEM, and data replay maintain high plasticity, whereas the regularization algorithms do not do so. (B) In the pre-readout activation space, the activation change is quantified as the difference between task 1 activations after training on task 1 and after training on all 11 tasks (displacement). With increasing regularization strength, the regularization algorithms reduce the activation change in both the range and null space of task 1 readout, which corresponds to the observed maintenance of stability and loss of plasticity. With increasing replay buffer size (and memory strength in the case of GEM), the rehearsal algorithms maintain high activation change in both the range and null space of the task 1 readout, which ensures plasticity. The use of non-linear, many-to-one mapping loss functions by these algorithms allows for changes in the range that are not detrimental to stability. These results illustrate the utility of the RDAC framework as a diagnostic tool to understand the stability-plasticity trade-offs of continual learning algorithms. The diagonal dotted line in panel B and the two insets indicates the expected ratio of displacement in null space and range of a readout if activations were to move uniformly in all dimensions of the pre-readout layer.

the network is first trained on CIFAR-10, and then sequentially trained on ten equal task splits from CIFAR-100 (we ran the analysis 3 times with randomly chosen assignments of classes to splits, and show the results averaged across those seeds). Images were subjected to augmentations. Separate readouts were trained for each of the tasks, with softmax activation and cross-entropy loss. After training on task 1 (CIFAR-10), we computed the projection matrices $\mathbf{CC}^\top$ and $\mathbf{NN}^\top$, using singular value decomposition (SVD).

## 4.2 RESULTS

We assessed the stability-plasticity trade-offs of the CL algorithms from the perspective of the RDAC framework. We considered the performance on task 1 (measure of stability) and on task 11 (measure of plasticity) after training on all 11 tasks. The changes (displacement) of the task 1 activations after task-1 training to after task-11 training, decomposed into the range and null space of the trained task 1 readouts, were assessed. Results are shown in Figure 2 (panels A & B).

First, in terms of desired behavior, all the algorithms gain stability, with increasing regularization strength for regularization algorithms, EWC, SI, and LwF, and with increasing replay buffer size (and memory strength in the case of GEM) for the rehearsal algorithms, GEM and data replay. However, there are differences between the regularization and rehearsal algorithms in how this stability is achieved. In the regularization algorithms, the activation changes in the range decreased with increasing stability (Case 1). In the rehearsal algorithms, the activation changes did not de-

crease similarly (Case 2). This is attributable to GEM and data replay being able to utilize the many-to-one, non-linear loss function (cross-entropy loss between the prediction and $1-$hot targets) to allow movement outside the effective range which is smaller than the readout range. Additionally, in GEM, the activation changes in the range increased with increasing stability. This is related to the backward transfer constraint - GEM encourages gradients for the new task to point towards the gradients estimated (with the replay buffer) for prior tasks, to jointly reduce the loss on all the tasks. Moving along task 1 gradients, which point into the task 1 readout range, leads to a change in the activations in the range.

Second, in terms of desired behavior, the rehearsal algorithms maintain plasticity with increasing replay buffer size (and memory strength in the case of GEM). The activation changes in the null space did not decrease substantially (Case 6). Additionally, in GEM, the activation changes in the null space increased while plasticity did not decrease substantially. As mentioned earlier, GEM encourages gradients to point into all the gradients estimated with the readouts until task $n-1$ while training on task $n$. The readouts of task 2 to task $n-1$ occupy some of the null space of the task 1 readout. Thus GEM also leads to activation change into the task 1 readout null space.

Third, in terms of undesired behavior, the regularization algorithms lose plasticity with increasing regularization strength. The activation changes in the null space decreased substantially (Case 7). In the case of EWC and SI, each weight in the network is assigned an importance score based on either the gradients for task $k$ post task $k$ training (used to estimate Fisher information in EWC) or the integration of gradients across task $k$ training (SI). During training on new tasks, a constraint is introduced - the weights during training on task $n$ should be similar to the weights post task $k$ training, weighted by the importance scores of those weights for that task, over all tasks $k < n$. This procedure assumes the gradients are sufficiently small such that this importance-based constraint ensures weight change into directions orthogonal to those that would affect the losses for the prior tasks. Viewing the results from our framework, we speculate that either the gradients are not small enough or the importance score computation is not sufficiently constrained to only span the union of the ranges of the prior readouts, thus over-constraining the null spaces of the prior readouts. Extensions to EWC and SI should allow for null space activation change in order to maintain plasticity.

The case of LwF is intriguing. LwF generates pseudo-labels (not $1-$hot) for the current task inputs on the prior readouts, before training on the task. During training, the algorithm enforces the constraint that the predictions onto the prior readouts should match the pseudo-labels. Doing so reduces the activation changes of the prior tasks' inputs in the prior readout range. As the constraint is imposed based on prior task readouts, we would expect that there is no pressure to restrict the activation changes in the null space. However, we observed a decrease in null space activations with increasing strength of the constraint. A possible explanation is that the range of the readout relies on a substantial part of the upstream network (i.e. discriminative features for a task rely on a lot of combinations of low-level features). In asking the range to be clamped, the gradients inadvertently end up clamping the space outside the range downstream, which is observed as null space clamping. How a regularization algorithm could get around this issue is an open problem. Such an algorithm would need to explicitly allow for interaction between the gradients related to the prior readout range and null space and the gradients related to the new task.

In sum, our results show that the principles laid out in the RDAC framework relating readout-decomposed activation changes to the stability-plasticity trade-offs provide useful insights into the inner workings of continual learning algorithms for deep, non-linear neural networks. These patterns of results were also found to hold in a larger dataset and architecture (see Appendix A.2), making our insights generalizable.

## 5 USING INSIGHTS FROM RDAC TOWARDS BUILDING A CONTINUAL LEARNING ALGORITHM

We seek to derive a continual learning algorithm, primarily to empirically validate the link between the readout decomposition of activation changes and the stability-plasticity trade-off. As a start, in one-hidden-layer linear neural networks, we show how the activation changes are linked to the weight updates, and analytically derive a gradient-decomposition-based algorithm to ensure maximal stability without further throttling plasticity.

The input-output mapping of a one-hidden-layer linear network, with no bias terms, can be written as $\mathbf{o} = \mathbf{x}\mathbf{W}_{\mathbf{H}}^{\top}\mathbf{W}_{\mathbf{R}}^{\top}$, where $\mathbf{o}^{1\times o}$ is the output with $o$ neurons, $\mathbf{x}^{1\times x}$ is the input with $x$ features, $\mathbf{W}_{\mathbf{H}}^{h\times x}$ is the mapping from input to the hidden layer with $h$ neurons, and $\mathbf{W}_{\mathbf{R}}^{o\times h}$ is the mapping from the hidden layer to the output. After training on task 1: $\{\mathbf{x}^{1} \rightarrow \mathbf{o}^{1}\}$, we get the trained readout $\mathbf{W}_{\mathbf{R}^{1}}$. While training on task 2: $\{\mathbf{x}^{2} \rightarrow \mathbf{o}^{2}\}$, we get the gradient $\mathbf{\Delta W_{H}}$. In order to maintain stability, we want the learned task 1: $\{\mathbf{x}^{1} \rightarrow \mathbf{o}^{1}\}$ mapping to stay preserved.

Given a projection matrix $\mathbf{A}^{h\times h}$ applied to the gradient, we want:

$$\mathbf{o^1} = \mathbf{x^1}(\mathbf{W_H} + \mathbf{A}\mathbf{\Delta W_H})^{\top}\mathbf{W_{R^1}^{\top}} \implies \mathbf{x^1}(\mathbf{A}\mathbf{\Delta W_H})^{\top}\mathbf{W_{R^1}^{\top}} = \mathbf{0} \implies \mathbf{W_{R^1}}\mathbf{A} = \mathbf{0} \quad (1)$$

One solution is $\mathbf{A} = \mathbf{N}\mathbf{N}^{\top}$, where $\mathbf{N}$ is the null space matrix of $\mathbf{W_{R^1}}$. Projecting the gradient for the $\mathbf{W_H}$ into the null space of $\mathbf{W_{R^1}}$ will ensure stability. Meanwhile, the hidden layer activations $\mathbf{h^1}$ are free to change in the null space of $\mathbf{W_{R^1}}$ which allows plasticity. This is our gradient decomposition algorithm for one hidden layer linear networks.

To allow full control over the stability-plasticity trade-off, we can consider $\mathbf{A} = \alpha\mathbf{C}\mathbf{C}^{\top} + \beta\mathbf{N}\mathbf{N}^{\top}$, where $\alpha$ (termed "range weight") and $\beta$ (termed "null space weight") are scalars that weigh the projections into range and null space of $\mathbf{W_{R^1}}$. $\mathbf{W_{R^1}}\mathbf{A} = \mathbf{0}$ only if the range weight $\alpha = 0$, which ensures stability. Stability does not rely on the null space weight $\beta$ but plasticity does as, if $\beta$ is reduced while $\alpha = 0$, the gradient becomes smaller, and learning the new task becomes slower. In Section 6, we empirically test this relationship between the range and null space weights, and the stability-plasticity trade-off.

Could we analytically derive a similar algorithm for deep, non-linear networks? The introduction of non-linearities and multiple hidden layers makes the procedure seen in Eq. 1 non-trivially complex. Backpropagation needs to be used to estimate the upstream gradients, and the simplest resulting solutions require the estimation of multiple null spaces at every iteration of training. This makes the implementation inefficient. The multiple hidden layer linear network case is analyzed in Appendix A.3. The non-linear case is analytically more complex to analyze, as the simple decomposition of the task mapping into the old mapping and the change in that mapping, as seen in Eq. 1, is not possible.

## 6 Assessing the gradient decomposition algorithm with RDAC

Using the RDAC framework, we present an assessment of the stability-plasticity trade-offs of the gradient decomposition algorithm. First, we outline the network architecture and task settings. Second, we present the results assessing our decomposed-activation-space-derived control over the stability-plasticity trade-off. Third, to understand the complexity of the network and task setting, we assess the stability-plasticity trade-offs of EWC in this setting. We chose EWC due to its popularity, its being a regularization algorithm, the relative ease of its implementation outside the Avalanche library, and because it has only one regularization parameter.

### 6.1 Network and task

We considered a neural network with one hidden layer with 11 neurons with no bias terms, which maps MNIST images (LeCun et al., 1998) to their classes (The exact architecture and optimization procedure is described in Appendix A.1.2). We considered the split-MNIST setting where the first task is to classify the first five digits (0-4), and the second task is to classify the last five digits (5-9). Images were subjected to augmentations. Separate readouts were trained for the two tasks, with softmax activation and cross-entropy loss. After training on task 1, we computed the projection matrices $\mathbf{C}\mathbf{C}^{\top}$ and $\mathbf{N}\mathbf{N}^{\top}$, using SVD.

### 6.2 Results

First, we assessed if the gradient decomposition algorithm managed to maintain stability without trading off plasticity, and whether varying the range and null space weights ($\alpha$ and $\beta$) led to the control over the stability-plasticity trade-off outlined in the RDAC framework. As seen in Figure 3A, when $\alpha = 0$ and $\beta = 1$, the performance on both task 1 (stability) and on task 2 (plasticity) was

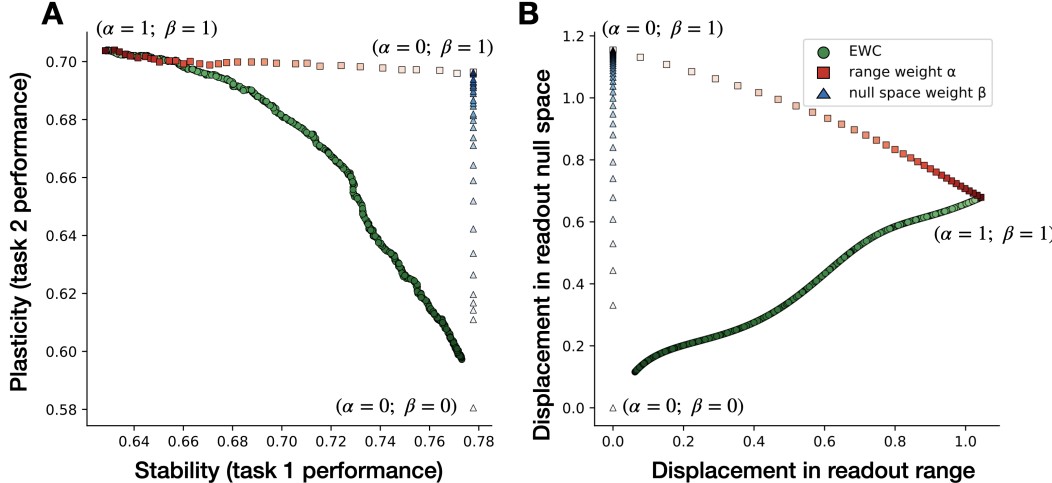

Figure 3: Continual learning in one hidden-layer linear neural networks learning to classify MNIST digits. (A) The gradient decomposition algorithm can maintain both stability and plasticity when the gradients are projected into the null space of the previous readout ($\alpha = 0$, $\beta = 1$). Allowing the gradients to be projected into the range of the readouts ($\alpha > 0$) is detrimental to stability, whereas restricting the gradients from being projected into the null space of the readouts ($\beta < 1$) is detrimental to plasticity. Additionally, Elastic weight consolidation (EWC) trades off plasticity for stability, indicating that this problem is non-trivial from a continual learning standpoint. Darker hues correspond to higher values of $\alpha$, $\beta$, and the EWC regularization strength $\lambda$. (B) In the hidden-layer activation space, the gradient decomposition algorithm effectively restricts changes in the activations for task 1 to the null space of the task 1 readout. Restricting the change in the null space is detrimental to plasticity, as seen in the case of EWC and when we set $\beta < 1$. These results illustrate the utility of the RDAC framework in informing the construction of new continual learning algorithms.

high. We also observed no task 1 activation change in the readout 1 range, but observed substantial change in the null space which corresponds to the network learning task 2 (Figure 3B). Increasing $\alpha$ to 1 increased the activation change in the readout range, thereby reducing the stability. On the other hand, lowering $\beta$ to 0 decreased the activation change in the readout null space, thereby reducing the plasticity. These simulations both show the efficacy of the gradient decomposition algorithm and confirm the control over the stability-plasticity trade-off outlined in the RDAC framework.

Next, to assess the complexity of the split-MNIST task in a one-hidden-layer linear network, we assessed the stability-plasticity trade-offs of EWC. If it is trivial to jointly maintain stability and plasticity, we reasoned that EWC can fare well on this task as compared to the task in the previous section. As seen in Figure 3A, increasing the regularization strength of EWC increased the stability at the expense of plasticity. Analyzing the activation change in the readout range and null space revealed that EWC ended up restricting change not only in the range (which is useful for stability) but also in the null space which was detrimental to plasticity (Figure 3B). This indicates that the network and task settings in use constitute a non-trivial continual learning problem.

These results confirm that the principles of the RDAC framework can be fruitfully put into practice. Although we do not derive a scalable continual learning algorithm in this work, the insights gained through the analysis of the one hidden layer neural network validate the RDAC link between the stability-plasticity trade-offs and the decomposition of activation changes. Future work needs to solve the challenge of explicitly disentangling the upstream projections of the range and null space of the prior readouts, to be able to fully establish the link between the decomposed activation changes and the stability-plasticity trade-off, in deep non-linear networks.

# 7 DISCUSSION

In this study, we introduced the Readout-Decomposition of Activation Change (RDAC) framework which provides a unique perspective on the stability-plasticity dilemma in continual learning, by viewing the continual learning problem from the standpoint of the readout layer. The core insight is as follows: Readouts span a subspace (the range) in activation space. Restricting learning from making changes to the projections of the activations of prior tasks to that subspace is sufficient to ensure stability. Further restricting any change to the activations in the subspace other than that spanned by the readout (the null space) is detrimental to plasticity. We analyzed popular continual learning algorithms and provided insights into the trade-offs these algorithms make in terms of stability and plasticity, and how they relate to changes in the activations for prior tasks from the perspective of the prior readouts. Additionally, given the insights from the framework, for one-hidden-layer linear networks, we derived a gradient decomposition algorithm that restricts learning-induced activation changes to only the null space of prior readouts to maintain plasticity while ensuring stability, further validating the RDAC link between the stability-plasticity trade-off and readout-decomposed activation changes.

An intriguing facet of our framework lies in its demonstration that activation changes within the null space of the readout are linked to learning. We demonstrated this relationship with the diagnostic analysis on EWC in Section 4.2, and the analysis of the gradient decomposition algorithm in Section 6. Now consider the following situation: an observer unaware of the continual learning algorithm observes the network activations for a fixed set of stimuli over learning. The activations would keep changing across time which could make the observer wonder how the network reads out of such a dynamic code. Moreover, in the case where the loss functions used by the algorithm are non-linear, the resultant degeneracy might make any readout trained by the observer at a time point harder to generalize to other time points. Such signatures of activation change and the optimal-readout changing over time have been observed in mammalian brains, and the activation change is termed "representational drift" (Driscoll et al., 2017). Recently it has been suggested that the reason drift exists is because animals are engaged in learning (Aitken et al., 2022; Masset et al., 2022; Driscoll et al., 2022; Micou & O'Leary, 2023). This suggestion dovetails with the RDAC framework under which, drift, especially in the null space of the readout, is essential for continuous adaptation through the lifetime of an organism.

The RDAC framework is designed with a focus on task-incremental learning, where new readouts are trained for each task encountered during the continual learning process, and the prior readouts are frozen. There exist other continual learning scenarios van de Ven et al. (2022), in which the readout layer can be learnable (Rebuffi et al., 2017) and be shared across multiple tasks with the addition of a context signal (Cheung et al., 2019). Allowing the readout to adapt to new tasks, or by including context-signals, enabling the network to distinguish between different tasks and appropriately adjust its responses, can increase the effective capacity that a learning algorithm can leverage, facilitating more plasticity and stability (Thorat et al., 2019; Hummos, 2022). However, it is not immediately clear how the RDAC framework could be extended to these scenarios, but accommodating them is essential for building a more versatile and adaptable framework for studying continual learning.

In closing, the RDAC framework primarily provides a diagnostic tool to shed light on the intricate dynamics of stability and plasticity. It also provides analytical insights for the future development of continual learning algorithms. The journey of continual learning research continues, with our work offering a valuable contribution towards achieving the delicate balance between preserving past knowledge and adapting to new challenges in an ever-evolving world of information.

## REPRODUCIBILITY STATEMENT

Details about the implementation of the continual learning algorithms are mentioned both in the Sections 4 and 6, and in the Appendix A.1. The gradient decomposition algorithm is derived and fully explained in Section 5.

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

# A    APPENDIX

## A.1    METHODS

### A.1.1    ASSESSING EXISTING CONTINUAL LEARNING ALGORITHMS WITH RDAC

For the experiments on the Cifar110 task, we construct 11 datasets (one for each task). The first task, on which we perform the bulk of our analyses consists of the full Cifar10 dataset (with usual training and validation splits). For each subsequent task, we sample 10 unique classes from Cifar100. Experiments are repeated with three different repeats of this procedure, providing some control for the varying difficulty of the different task splits. Data for all tasks was augmented with random cropping (padding $= 4$) and horizontal flipping throughout training. All data was normalized with means $(0.5071, 0.4865, 0.4409)$ and standard deviations $(0.2673, 0.2564, 0.2762)$ for the RGB channels.

All networks were trained with Adam ($lr = 0.001$, $\beta_1 = 0.9$, $\beta_2 = 0.999$) for 60 epochs per task. Following the findings in Li & Hoiem (2017), we warm up the new readout at the start of each new task (excluding training on the first task). This has been reported to stabilize representations at the start of training on a new task (where the randomly initialized new readout is not aligned with the features of the remainder of the network, causing large gradients). We freeze the weights of all layers except the new readout for the first 10 epochs of training. Additionally, since our analyses investigate activation changes relative to the range of previously learned readouts, we freeze all parameters in old readouts for methods that would otherwise allow changing readout weights for old tasks (this is the case for LwF and data replay).

The network architecture for these experiments is adopted from Zenke et al. (2017) and has been slightly altered. It consists of two VGG blocks (32 channels in the first, 64 channels in the second block each, kernel size 3). Each block of two convolutional layers is followed by a max pool layer with kernel size and stride 2. The pre-readout dense layer was scaled to have 128 output units and no dropout was used throughout the network. All layers in the backbone were initialized with Kaiming-He He et al. (2015) initialization as implemented in PyTorch.

After performing initial sweeps for the hyperparameters in the tested algorithms to determine the rough effective ranges, we performed additional sweeps for each algorithm in order to generate the data points in Figure 2. Each data point visualized is the average over three experiments with the same hyperparameter settings, but different seeds (and therefore task splits as described above).

Hyperparameters were swept as follows:

- For EWC, $\lambda$ was varied between $10^{-1}$ - $10^5$.
- For SI, $\epsilon$ was fixed to 1 and $\lambda$ was varied between $10^{-2}$ and $10^5$.
- For LwF, we fixed the temperature to 1 and varied alpha between $0.01$ and $10$.
- For data replay, we used replay buffer sizes between 0 and 60000 samples, with the default replay buffer style as implemented in Avalanche (as of version 0.3.1).
- For GEM, we varied the memory strength parameter ($\gamma$ in the original publication) between 0 and 1 and varied the number of patterns stored per experience to estimate the gradient projection between 0 and 20000.

### A.1.2    ASSESSING THE GRADIENT DECOMPOSITION ALGORITHM WITH RDAC

The linear system described in section 6 is a one-hidden layer network without biases and 11 units in its hidden layer. The network has two separate linear readouts with 5 units each, to accommodate the split MNIST task. For all experiments, the network was trained for 30 epochs per task, with plain stochastic gradient descent and a learning rate $5.10^{-4}$ and batch size 16.

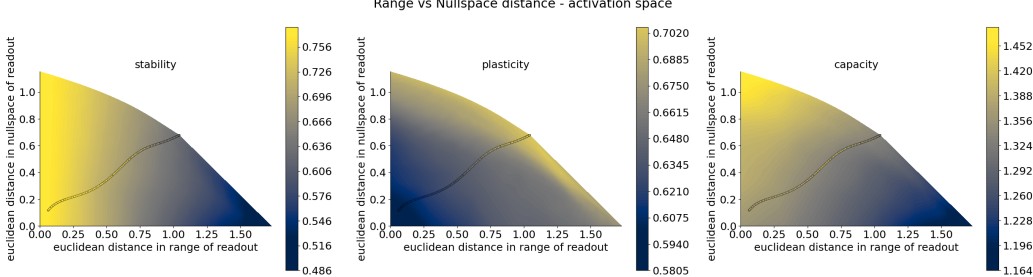

Figure 4: Movement in range and null space for gradient decomposition (surface) and EWC (scatter points). Points and contours are coloured by plasticity, stability and "capacity" computed as plasticity + stability.

Since the Split MNIST task is very easy, even for a small linear network we increase the difficulty of the dataset slightly by applying a number of transformations to the dataset once at the time of constructing the dataset. This increases the effect of catastrophic forgetting while keeping a fixed dataset, allowing for easy experimentation. The transformations were implemented using the torchvision transforms package. Images of digits were augmented with random rotations ($\pm 10$ degrees), translations ($\pm 10$ percent of image size in both axes), scaled between 90-110% of the original size and randomly cropped with padding = 4. Finally, we applied the 'ColorJitter' transformation with parameters brightness = 0.1, contrast = 0.1, saturation = 0.1, and hue = 0.1. Transformations are only applied to training data for both tasks.

For EWC, we approximate the diagonal of the Fisher information matrix for the hidden layer parameters as the square of the gradients for the first task over the whole dataset for task 1.

$$F_w = \frac{\sum_N (\Delta w)^2}{Nb}$$

,

for N batches of data (with $b$ samples each). We sweep 1000 values for the scalar multiplier $\lambda$ governing regularization strength on a log scale between 0 and $10^5$.

To illustrate our gradient decomposition result, we swept the space of possible decompositions in a grid with 33 linear spaced values between 0 and 1 for $\alpha$ and $\beta$. In Figure 3 we visualized the extremes of this search, and the results of the full space are included in Figure 4 for completeness.

## A.2  RDAC ASSESSMENT OF CONTINUAL LEARNING ON A LARGER NETWORK AND DATASET

To assess whether our findings on the Cifar110 scale to larger tasks and stimuli, we repeat our analysis of EWC and data replay (previous readouts frozen) on the TinyImagenet dataset Le & Yang (2015), adapting the slimmed ResNet18 as reported in Lopez-Paz & Ranzato (2017) and implemented in Avalanche Carta et al. (2023). TinyImagenet consists of 64x64x3 images belonging to 200 classes, which we split into 5 unique subsets of 40 classes per task.

As before, we sweep over regularisation strengths for both EWC and Replay. For EWC, we sweep lambda in [1 .. 100000], for Replay we vary replay buffer sizes in [1000 .. 50000]. To allow for estimation of movement in the nullspace and range of previous readouts, we freeze all parameters in the readout layers for previously trained tasks.

All networks were trained for 60 epochs per task, using the Adam optimizer with the same settings as used for our earlier experiment. For tasks 2 to 5, we only train the new readout for the first 10 epochs, to align the new readout with the rest of the network before propagating gradients.

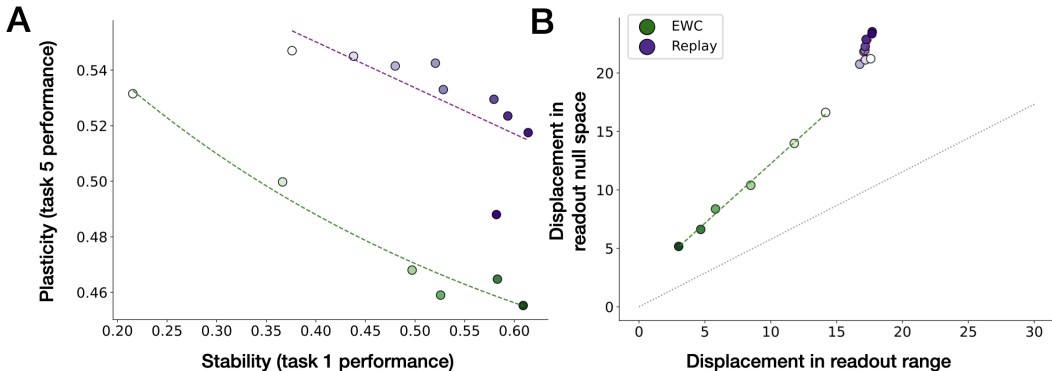

Figure 5: Stability-Plasticity tradeoff as observed after training on 5 splits of the TinyImagenet dataset. (A) Displacement of representations of the first task after training on all 5 tasks, decomposed into nullspace and range movement using our readout decomposition method. (B). Analogous to the results observed in Cifar110, we see that replay can achieve high stability while sacrificing less plasticity compared to the regularisation-based method EWC. As observed before, higher plasticity observed in replay correlates with more movement of previous task representations. EWC strongly restricts the movement of previously learned representations and cannot maintain high plasticity as regularisation strength increases.

## A.3 CONTINUAL LEARNING IN A THREE-LAYER LINEAR NEURAL NETWORK

In order to demonstrate the complexity of deriving an efficient gradient decomposition algorithm (cf. Section 5) for multi-layer linear networks, we consider the case of a three-hidden layer linear network: $\mathbf{o} = \mathbf{x}\mathbf{W}_{\mathbf{H_1}}^\top \mathbf{W}_{\mathbf{H_2}}^\top \mathbf{W}_{\mathbf{H_3}}^\top \mathbf{W}_{\mathbf{R}}^\top$.

After training on task 1: $\{\mathbf{x^1} \to \mathbf{o^1}\}$, we get the trained readout $\mathbf{W_{R^1}}$. While training on task 2: $\{\mathbf{x^2} \to \mathbf{o^2}\}$, we get the gradient $\mathbf{\Delta W_H}$. In order to maintain stability, we want the learned task 1: $\{\mathbf{x^1} \to \mathbf{o^1}\}$ mapping to stay preserved.

We want: $\mathbf{o^1} = \mathbf{x^1}(\mathbf{W_{H_1}} + \mathbf{\Delta W_{H_1}})^\top (\mathbf{W_{H_2}} + \mathbf{\Delta W_{H_2}})^\top (\mathbf{W_{H_3}} + \mathbf{\Delta W_{H_3}})^\top \mathbf{W_{R^1}^\top}$, implying:

$$\mathbf{x^1}(\mathbf{\Delta W_{H_1}^\top} \mathbf{W_{H_2}^\top} \mathbf{W_{H_3}^\top} + \mathbf{W_{H_1}^\top} \mathbf{\Delta W_{H_2}^\top} \mathbf{W_{H_3}^\top} + \mathbf{W_{H_1}^\top} \mathbf{W_{H_2}^\top} \mathbf{\Delta W_{H_3}^\top} + \mathbf{W_{H_1}^\top} \mathbf{\Delta W_{H_2}^\top} \mathbf{\Delta W_{H_3}^\top} +$$
$$\mathbf{\Delta W_{H_1}^\top} \mathbf{W_{H_2}^\top} \mathbf{\Delta W_{H_3}^\top} + \mathbf{\Delta W_{H_1}^\top} \mathbf{\Delta W_{H_2}^\top} \mathbf{W_{H_3}^\top} + \mathbf{\Delta W_{H_1}^\top} \mathbf{\Delta W_{H_2}^\top} \mathbf{\Delta W_{H_3}^\top})\mathbf{W_{R^1}^\top} = \mathbf{0} \qquad (2)$$

There could be multiple ways of constraining the gradients to satisfy Eq. 2. However, we explored if, parsimoniously, a solution could only constrain the pre-readout gradient $\Delta W_{H_3}$ and let backpropagation take care of the rest.

Using backpropagation we can write out the gradients in terms of the derivative of the loss function, $\mathbf{e_o} = \frac{\partial \mathcal{L}(\mathbf{o},\mathbf{o^2})}{\partial \mathbf{o}}$ (assuming batch size 1 here, mapping $\mathbf{x^2} \to \mathbf{o^2}$). $\mathbf{e_o}$, and not $\Delta W_{H_3}$, is propagated back for upstream gradient computations as it is independent of the network activations. The gradients are computed as follows:

$$\mathbf{\Delta W_{H_1}^\top} = (\mathbf{x^2})^\top \mathbf{e_o} \mathbf{W_{R^2}} \mathbf{W_{H_3}} \mathbf{W_{H_2}}$$
$$\mathbf{\Delta W_{H_2}^\top} = \mathbf{W_{H_1}}(\mathbf{x^2})^\top \mathbf{e_o} \mathbf{W_{R^2}} \mathbf{W_{H_3}}$$
$$\mathbf{\Delta W_{H_3}^\top} = \mathbf{W_{H_2}} \mathbf{W_{H_1}}(\mathbf{x^2})^\top \mathbf{e_o} \mathbf{W_{R^2}}$$

We would like to know the transformation $\mathbf{e_o} \to \mathbf{A}\mathbf{e_o}$ which satisfies the constraint in Eq. 2, when this transformed $\mathbf{e_o}$ is backpropagated. As a first step, we can ignore the interactions between the terms of Eq. 2 by asking them to be independently 0. The resulting transformation is jointly subject to 3 constraints:

1. In order to maintain a non-zero gradient $\mathbf{\Delta W_{H_3}^\top}$, while zeroing the terms associated with it in Eq. 2, $\mathbf{A}$ should project gradients into the null space of $\mathbf{W_{R^1}} \mathbf{W_{R^2}^\top}$

2. In order to maintain a non-zero gradient $\mathbf{\Delta W}_{\mathbf{H_2}}^{\top}$, while zeroing the remaining terms associated with it in Eq. 2, $\mathbf{A}$ should also project gradients into the null space of $\mathbf{W_{R^1}} \mathbf{W_{H_3}} \mathbf{W}_{\mathbf{H_3}}^{\top} \mathbf{W}_{\mathbf{R^2}}^{\top}$

3. In order to maintain a non-zero gradient $\mathbf{\Delta W}_{\mathbf{H_1}}^{\top}$, while zeroing the remaining term associated with it in Eq. 2, $\mathbf{A}$ should also project gradients into the null space of $\mathbf{W_{R^1}} \mathbf{W_{H_3}} \mathbf{W_{H_2}} \mathbf{W}_{\mathbf{H_2}}^{\top} \mathbf{W}_{\mathbf{H_3}}^{\top} \mathbf{W}_{\mathbf{R^2}}^{\top}$

Intuitively, this is similar to canceling the propagation of $\mathbf{e_o}$ into readout 1 (through all of the 3 paths listed above) i.e. any errors that would be induced in readout 1 by changing any of the weights would become zero due to such a projection, $\mathbf{Ae_o}$. This algorithm would ensure stability, however, it is unclear how much plasticity can be leveraged after the said projection - if the intersection of the null spaces spans a very low-dimensional space, plasticity will be hampered. This needs to be studied empirically for a variety of datasets.

This parsimonious algorithm is computationally expensive as compared to the gradient decomposition algorithm for the one-hidden layer network discussed in Section 5. In the current algorithm, during every weight update, the 3 null spaces need to be computed, as the weights keep changing. Additionally, the number of null spaces to be computed scales with the number of hidden layers $n$ and with the number of tasks $k$, as $n(k-1)$. With increasing $n$ and $k$, the intersections of the null spaces would get smaller and plasticity would be hampered. How much of this is a problem for existing continual learning datasets needs to be tested empirically.

