# OpenReview forum: "Balancing Stability and Plasticity in Continual Learning: the readout-decomposition of activation change (RDAC) framework"
_ICLR.cc/2024/Conference — Submitted to ICLR 2024_

### Official Review · Reviewer_djfe · 2023-10-27

**Soundness:** 3 good
**Presentation:** 3 good
**Contribution:** 3 good
**Rating:** 6
**Confidence:** 1

**Summary:**

This paper considered balancing the learning forgetting trade-off in continual learning. Specifically, this paper considered the changement of representation \delat_h into two parts, the range of readout and Null space of readout space (maintains stability).  Then this paper derived a gradient decomposition algorithm to explicitly control the learning-forgetting trade-off. Empirical results demonstrated improved trade-off.

-----Post-rebuttal update
I would appreciate authors' efforts, which addressed my concerns. I would maintain my current evaluation.

**Strengths:**

**Disclaimer: I did not work on continual learning and thus I could not evaluate the novelty/significance parts of this paper.**  Below is my evaluation from a general view.

I would think this paper presents a clear and interesting analysis in the learning/forgetting trade-off in continual learning, which seems quite important in continual learning. I would agree with the authors with their analysis on the representation decomposition (Fig 1).

The experimental parts clearly demonstrated the benefits of such an analysis by a better trade-off.

**Weaknesses:**

I would think the clarity part in gradient decomposition could be better elaborated. I would think a clear algorithm should be presented to show how to explicitly control the learning/forgetting trade-off.

**Questions:**

See the weakness parts.

---

> ### Author Response · Authors · 2023-11-20
> **Rebuttal**
>
> Dear reviewer,
>
> Thank you for your comments. Please check the comment - https://openreview.net/forum?id=9vkgAaCI3F&noteId=rcFk2d3LLJ - for a summary of the changes to the paper.
>
> We have taken your suggestion into account and reworded that section, Section 5, to make it clear what parts refer to the gradient decomposition algorithm and what refers to the causal analysis between the stability-plasticity trade-off and the activation changes in the readout range and null space.
>
> We hope that our responses and revisions have clarified our contributions and we want to thank you for your helpful insights.
>
> Thank you, The Authors

---

### Official Review · Reviewer_gbcF · 2023-10-29

**Soundness:** 2 fair
**Presentation:** 3 good
**Contribution:** 2 fair
**Rating:** 5
**Confidence:** 5

**Summary:**

This paper introduces the Readout-Decomposition of Activation Change (RDAC) framework, which aims to address the stability-plasticity trade-off in continual learning (CL) algorithms. This trade-off is a significant challenge in preserving prior information while acquiring new knowledge. The RDAC framework dissects this trade-off by linking learning-induced activation changes to stability and plasticity, thereby offering insights into CL algorithms.

Moreover, This paper presents a gradient decomposition algorithm for one-hidden-layer linear neural networks. This algorithm restricts activation changes within the range of prior readouts, maintaining stability without significant loss of plasticity.

The RDAC framework sheds light on the connection between learning-induced activation changes and the stability-plasticity trade-off, providing insights into representational drift in biological systems. The results show that GEM and data replay preserved stability and plasticity, while SI, EWC, and LwF traded off plasticity for stability.

**Strengths:**

The Readout-Decomposition of Activation Change (RDAC) framework addresses the stability-plasticity dilemma and its relation to catastrophic forgetting, and relates learning-induced activation changes to the degree of stability and plasticity. The paper contributes to ongoing efforts in understanding and solving the complexities of continual learning.


The paper also presents a gradient decomposition algorithm for one-hidden-layer linear neural networks to maintain high stability without sacrificing plasticity. Overall, the RDAC framework provides valuable insights into CL algorithms and offers potential for novel CL approaches.

**Weaknesses:**

Theoretical analyses on one-hidden-layer linear neural networks are not scalable or too instructive.

This paper lacks empirical results, comparisons, and practical implications, which could limit its applicability in real-world scenarios.

The paper focuses primarily on evaluating existing CL algorithms and does not propose any new algorithms or techniques.

Lack of code and hyper-parameter configuration.

**Questions:**

Have you considered evaluating the RDAC framework on other CL datasets or tasks? It would be interesting to see how the framework performs in different settings and if the results hold across a broader range of scenarios.

It would be helpful to have more empirical results and comparisons with other frameworks or approaches to validate the effectiveness of the RDAC framework. Are there any plans to conduct such experiments in the future?

How generalizable is the gradient decomposition algorithm for linear neural networks? Can it be extended to more complex network architectures, such as deep neural networks, and still maintain stability without sacrificing plasticity?

---

> ### Author Response · Authors · 2023-11-20
> **Rebuttal**
>
> Dear reviewer,
>
> The paper has substantially improved due to your insights - thank you. Please check the comment - https://openreview.net/forum?id=9vkgAaCI3F&noteId=rcFk2d3LLJ - for a summary of the changes to the paper. Here we will address your comments.
>
> Re: Weakness, Point 1 (+ Questions, Q3):
> Analysis of the one-hidden layer neural network provides us with a causal link between the stability-plasticity trade-off and the activation changes in the readout range and null space. This is hard to do in deep non-linear networks as such a causal link is hard to derive analytically as shown in Appendix A.3. However, the approach shown in the linear case i.e. decomposition of Eq. 1 might provide a good starting point for subsequent attempts in deriving solutions.
>
> Re: Weakness, Point 2 (+ Questions, Q1):
> In Appendix A.2. we now show that our insights generalize to a larger network (ResNet) and dataset (TinyImagenet), making our conclusions more generally applicable.
>
> Re: Weakness, Point 3:
> As stressed in the revised Introduction, the goal of this paper is to present a new framework for analyzing continual learning algorithms, borne out of recent results mentioned in the Introduction. The gradient decomposition algorithm was primarily derived to verify the causal link between the stability-plasticity trade-off and the activation changes in the readout range and null space. Deriving such algorithms analytically for deep non-linear networks proves to be challenging (see Appendix A.3)
>
> Re: Weakness, Point 4:
> Code is now included as a .zip file. The most relevant hyperparameter settings are mentioned in the Appendix.
>
> Re: Questions, Q2:
> This suggestion dovetails with the suggestion made by Reviewer RvDh (Questions, Q5). In general, we agree that links between RDAC and other frameworks such PCA, information theory, information bottleneck theory, and rate-distortion theory would be interesting to explore in future research, as this lies beyond the scope of the current study.
>
> We hope that our responses and revisions have clarified our contributions and we want to thank you for your helpful insights.
>
> Thank you, The Authors

---

### Official Review · Reviewer_RvDh · 2023-10-31

**Soundness:** 2 fair
**Presentation:** 2 fair
**Contribution:** 3 good
**Rating:** 5
**Confidence:** 5

**Summary:**

The paper introduces the Readout-Decomposition of Activation Change (RDAC) framework to analyze the stability-plasticity tradeoff in continual learning algorithms. Readout layer is similar to the final linear probing layers in continual-learning/SSL scenarios. They perform SVD decomposition to get the range and null spaces. The main idea is that the range lets us observe stability, the more changes in that range, the less stability we have. On the other hand, changes in the null space allow plasticity for learning new tasks. Regularization methods restrict changes in both the range and null space, sacrificing plasticity for stability. In contrast, replay methods allow changes in the null space, maintaining plasticity.
Surprisingly, it may seem like replay-based methods should exhibit substantial catastrophic forgetting, since they allow significant changes to activations in the range of prior readouts. However, the paper shows this is not the case - these methods can maintain strong stability and plasticity. Finally, for a simple linear network, they derive a gradient decomposition algorithm that projects weight updates into the null space to maximize stability without reducing plasticity.

**Strengths:**

- It presents a novel perspective on analyzing continual learning through the lens of readout weights and their null spaces. This provides a new tool for disentangling stability vs plasticity.

- The gradient decomposition algorithm demonstrates a concrete application of the concepts that maintains stability and plasticity. I believe this gradient update method to be the biggest potential contribution of this paper.

- The paper is clearly written and does a good job explaining and visualizing the key ideas.

**Weaknesses:**

Authors acknowledged some of these limitations.

- So the explanatory power of the framework for nonlinear networks is unclear. The analysis relies on precisely computing the readout null space, which may be difficult in nonlinear models where the spaces are less well-defined. It's also unclear if the insights will fully translate when there are multiple nonlinear layers. The analysis of activation changes is quite limited for the nonlinear network experiments. They only approximate the null spaces for complicated models, analyze a single layer rather than the whole network, and observe overall trends without an in-depth study like was done for the linear case.

- The continual learning scenarios are relatively simple (Split CIFAR and MNIST). Needs more exploration on complicated dataset benchmarks and continual learning scenarios.

- No comparison against latest SOTA.

**Questions:**

If the authors are able to answer most of the questions and are able to further develop their algorithms as requested, this paper could significantly improve. This paper could have potential. The questions are very closely related to the weakness.

- 1. You mention that your work is related to representational drift in biological networks. More details on the biological connections and plausibility of the concepts will be helpful.

- 2. Do you have ideas for extending gradient decomposition algorithms to deep nonlinear networks?

- 3. You propose the readout null space allows plasticity for new tasks. But how can you formally quantify or guarantee the capacity for plasticity? (Aka I want to see more maths justifying that the null space gives us plasticity. Information theory perspectives or more experimental results might help).

- 4. Any performance guarantees of your proposed optimization algorithm?

- 5. Please comment on any links to PCA, information theory, information bottle neck theory, rate-distortion theory etc. Some helpful pointers:

  - a. PCA

    - i. PCA finds the principal components that capture the directions of maximum variance in a dataset. The readout range identified in this paper spans the subspace aligned with the readout weights. So both are identifying meaningful linear subspaces in high-dimensional data.

    - ii. The readout null space identified is analogous to the null space in PCA - directions that have no variance or are unimportant for the purposes of reconstruction/readout.

  - b. Rate-distortion theory:

    - i. Replay methods allow greater changes in the null space (less compression), maintaining plasticity. Regularization methods over-compress, losing plasticity.

---

> ### Author Response · Authors · 2023-11-20
> **Rebuttal**
>
> Dear reviewer,
>
> The paper has substantially improved due to your insights - thank you. Please check the comment - https://openreview.net/forum?id=9vkgAaCI3F&noteId=rcFk2d3LLJ - for a summary of the changes to the paper. Here we will address your comments.
>
> Re: Weakness, Point 1:
> Re: the point about "analyze a single layer rather than the whole network", the RDAC framework was built around the observation that readout misalignment constitutes a major problem in continual learning. This implies that we can get a lot of insights by just looking at the readout layer. As shown in our experiments, this turns out to be quite informative (Figures 2 & 3).
> Re: the point about "observe overall trends without an in-depth study like was done for the linear case", as we mentioned in Section 5 and Appendix A.3, it is hard to analytically derive a solution to an analog of Eq. 1 (Eq. 2 for a 3-hidden layer network) in deep non-linear networks. Moreover, the framework serves as a diagnostic tool for continual learning algorithms and the hope is that the analysis shown in Figure. 2 and outlined in Section 3 can be used generally (it works for a larger dataset and architecture as shown in Appendix A.2).
>
> Re: Weakness, Point 2:
> We now include results on continual learning on TinyImagenet with a slim ResNet that qualitatively shows the same results as in Figure. 2, solidifying the generality of our framework.
>
> Re: Weakness, Point 3:
> Our aim was to analyze continual learning algorithms from the two major classes - regularization and rehearsal, to assess our framework. All the algorithms considered are still competitive benchmarks in the field. Future work, especially when it comes to designing algorithms based on this framework, will indeed account for the SOTA algorithms.
>
> Re: Questions, Point 1:
> We are excited about the connections of this work to representational drift, however, given the page limitations, expanding on that relationship is outside the scope of this study.
>
> Re: Questions, Point 2:
> We have outlined our intuitions on extending that algorithm to deep networks in Appendix A.3. Currently, it poses a significant challenge but we hope our intuition paves the way.
>
> Re: Questions, Point 3:
> In the linear network case presented in Section 5, we show how we are guaranteed to maintain stability by projecting the gradient into the null space of the prior readout. Although we do not state formal guarantees for plasticity, what we can say is if the range of the readout spans a subspace - which it usually does as in task-incremental learning there are more neurons in the representational layer than readout classes - the null space is substantial and can be utilized by another task, fostering plasticity.
>
> Re: Questions, Point 4:
> The gradient decomposition algorithm guarantees stability as shown in Section 5, as a projection into the null space of the prior readout ensures that the already-learned mappings do not change (Eq. 1).
>
> Re: Questions, Point 5:
> Re: PCA - indeed, the PCs capture variance in the dataset. However, as mentioned in Section 2, the representations most probably span a larger space than the readout range.
> In general, we agree that links between RDAC and PCA, information theory, information bottleneck theory, and rate-distortion theory would be interesting to explore in future research as this lies beyond the scope of the current study.
>
> We hope that our responses and revisions have clarified our contributions and we want to thank you for your helpful insights.
>
> Thank you, The Authors

---

> > ### Comment · Reviewer_RvDh · 2023-11-22
> >
> > I have read the response, and also the other reviews.
> >
> > I thank the Authors for the effort. The answers are helpful, and also the improvement are useful. On the other hand, I feel that the major weaknesses remain and I will confirm my rating.

---

### Official Review · Reviewer_a3Rp · 2023-11-03

**Soundness:** 3 good
**Presentation:** 3 good
**Contribution:** 2 fair
**Rating:** 5
**Confidence:** 2

**Summary:**

The paper introduces the Readout-Decomposition of Activation Change (RDAC) framework to analyse plasticity and stability in networks performing continual learning. The framework projects the change in network activations upto the readout layer, onto the range and null space of the readout weights. The paper then hypothesizes that changes in the range-space projection between tasks represent changes in stability while changes in the null-space represent learning without changes in stability.

**Strengths:**

1. The paper is clearly written for the most part, and fairly easy to follow.
2. The RDAC framework is a neat idea to examine changes in the network weights / activations, and to analyse the stability-plasticity tradeoff during learning. The framework is simple enough that it can be applied to a variety of network architectures and tasks easily.

**Weaknesses:**

1. To the best of my understanding, the RDAC framework _interprets_ projections onto the null and range spaces and any changes thereof as representing stability and plasticity in the networks. This seems to be a logical leap that is not verified -- the experiments in section 4 and 5 simply analyse the changes of gradient projections for different networks in these spaces. While I find the connection between the various network hyperparameters and the changes in these spaces interesting, it is not clear whether they truly represent a stability / plasticity tradeoff without experiments also showing how changes in these subspaces are correlated with performance on the continual learning tasks.

2. The linear approximation in the RDAC framework is useful and makes it easy to analyse / apply to a variety of networks. However, given that it is non-trivial to derive a similar framework for non-linear activations or readouts, it is difficult to see how insights from the current linear framework can be extrapolated to the nonlinear setting. While the paper presents results on nonlinear networks in section 4, it is not clear whether these insights will extrapolate to other nonlinear networks.

Furthermore, there are no experiments showing how tuning the regularisation strength for SI, EWC and LwF, or memory strength / replay-buffer size for GEM to achieve a particular balance of stability and plasticity affects network performance in these tasks. This also makes it hard to judge whether the insights derived from the RDAC framework on nonlinear tasks really hold, can be extrapolated to other networks / methods and whether they are useful in developing methods for continual learning.

3. Related to point 2, while it is non-trivial to derive results for nonlinear projections, it would be good to have at least some intuition on how incorrect / applicable insights from a linear approximation would be to nonlinear networks, and whether any future endeavour to derive results for nonlinear networks could correct them.

**Questions:**

1. Do the changes in the range / null space truly correlate with plasticity / stability in training the networks?

2. How do we interpret insights from a linear approximation of gradient projections from a nonlinear network?

3. How bad are these linear approximations, and how can we correct them?

---

> ### Author Response · Authors · 2023-11-20
> **Rebuttal**
>
> Dear reviewer,
>
> The paper has substantially improved due to your insights - thank you. Please check the comment - https://openreview.net/forum?id=9vkgAaCI3F&noteId=rcFk2d3LLJ - for a summary of the changes to the paper. Here we will address your comments.
>
> Re: Weakness, Point 1 (+ Questions, Q1):
> In the updated paper, as described in Section 3, the predicted correlations between the stability-plasticity trade-off and the activation changes in the readout range and null space are complex. We laid out the different scenarios (Cases). In the results, Section 4.2., we link the behavior of various continual learning algorithms to those Cases, thereby establishing the predicted correlations. Indeed, this does not establish a causal link. However, in Section 5, we analytically derive such a causal link for a one-hidden layer linear network, which provides support to our account. Such a causal link is hard to establish for deep non-linear networks as we do not yet have a method for disentangling the projections of the readout range and null space in the upstream layers of the network, as mentioned in Section 5. In sum, we outlined a framework based on previous findings mentioned in the introduction and took the first steps in ascertaining its causal validity.
>
> Re: Weakness, Point 2:
> Indeed deriving an analytical solution for deep networks is hard, as now shown in Appendix A.3. However, the idea that the range is related to stability and that the null space is related to plasticity comes from considerations of activation changes at the readout, as explained in Section 3. Although we do not have an analytical solution for the deep non-linear case, we think that our intuitions are valuable in characterizing the link between the stability-plasticity trade-off and the activation changes in the readout range and null space
>
> Re: Weakness, point "Furthermore, there are no experiments showing how tuning the regularisation strength ...":
> Figure 2 shows the relationship between the regularization strength of each algorithm and its influence on the stability-plasticity trade-off and the readout-decomposed activation changes. To show that these results are generalizable, in Appendix A.2, we show that the same pattern holds for a larger dataset (TinyImagenet) and network (slim ResNet), giving us confidence that the framework is indeed generalizable.
>
> Re: Weakness, Point 3:
> in Appendix A.3, we now provide intuitions on the complexity of analytically deriving continual learning solutions for deep networks.
>
> Re: Questions, Q2 (+ Q3):
> None of our experiments engage in a linear approximation of gradient projections from a nonlinear network. In the linear network, we do project the gradients during learning. However, that projection is derived from the actual projection our framework wants which is related to the activations, see: Eq. 1. In the non-linear case, solving Eq. 1 does not involve simply projecting the gradients. As mentioned at the end of Section 5, "The non-linear case is analytically more complex to analyze, as the simple decomposition of the task mapping into the old mapping and the change in that mapping, as seen in Eq. 1, is not possible."
>
> We hope that our responses and revisions have clarified our contributions and we want to thank you for your helpful insights.
>
> Thank you, The Authors

---

### Author Response · Authors · 2023-11-20
**Overall update on the revision**

Dear reviewers,

Thanks to your comments and suggestions, the paper has improved substantially. We re-wrote major chunks of it and included new analyses. Here we mention the salient aspects of the revision.

1. In Section 1, the Introduction, we clarify that the goal of this paper is to outline a diagnostic framework for continual learning.
2. Section 3 has been updated to include a list of "Cases" that outline how stability and plasticity are predicted to be related to the activation changes in the range and null space. We link the behavior of the algorithms analyzed back to these Cases in order to properly substantiate our findings. Also, the derivation of the gradient decomposition algorithm has been pushed back to Section 5 to improve the flow of arguments.
3. Section 4.2 has been updated with a focus on the Cases listed in Section 3. Additionally, sections of the previous Discussion are now included here as they help us better explain the causes of the changes in activation space.
4. Section 5 now describes our algorithm for the single hidden layer linear network. We motivate the derivation by stating that we need an algorithm to establish the causal link between stability and plasticity and the readout-decomposed activation changes.
5. Section 7, the Discussion, has now been shortened to only summarize the already-discussed results and describe the link to representational drift and the limitations of our framework.
6. Appendix A.2 shows the results of our analysis, similar to the one in Section 4, during continual learning on TinyImagenet with a slim ResNet. These results indicate the generalizability of our framework.
7. In Appendix A.3, we analyze the case of a 3-hidden-layer linear neural network to show how complex it is to analytically derive a continual learning algorithm. Nonetheless, we hope the analysis provides useful pointers for designers of future continual learning algorithms.

We hope that this revision clarifies our contribution to continual learning and makes the paper a useful contribution to the ICLR readership.

Thank you,
The Authors

---

### Meta-Review · Area_Chair_FBff · 2023-12-18

**Metareview:**

This paper proposes the Readout-Decomposition of Activation Change (RDAC) framework to analyse the plasticity-stability tradeoff in continual learning. The main contribution of the work is the theory in linear case and analysis in one hidden layer neural network. While the authors do show ResNet results during the rebuttal, all the reviewers are concerned about the applicability of the analysis to complex non-linear settings. As the authors themselves agree, it is not easy to come up with a theory for the non-linear setting. However, to compensate for that, authors need to test their framework in a large number of continual learning tasks. Currently, the experiments are weak and the extension from linear to non-linear case is not convincing. I encourage the authors to add more tasks and also argue why the analysis would hold in non-linear settings as well.

**Justification For Why Not Higher Score:**

Both theory and experiment are weak.

**Justification For Why Not Lower Score:**

N/A.

---

### Decision · Program_Chairs · 2024-01-16

Reject